# Integrated Genomic Analysis of Primary Prostate Tumor Foci and Corresponding Lymph Node Metastases Identifies Mutations and Pathways Associated with Metastasis

**DOI:** 10.3390/cancers15235671

**Published:** 2023-11-30

**Authors:** Carlos S. Moreno, Cynthia L. Winham, Mehrdad Alemozaffar, Emma R. Klein, Ismaheel O. Lawal, Olayinka A. Abiodun-Ojo, Dattatraya Patil, Benjamin G. Barwick, Yijian Huang, David M. Schuster, Martin G. Sanda, Adeboye O. Osunkoya

**Affiliations:** 1Department of Pathology and Laboratory Medicine, Emory University, Atlanta, GA 30322, USA; cynthia.leigh.winham@emory.edu (C.L.W.); adeboye.osunkoya@emory.edu (A.O.O.); 2Department of Biomedical Informatics, Emory University, Atlanta, GA 30322, USA; 3Department of Urology, Emory University, Atlanta, GA 30322, USAdattatraya.patil@emory.edu (D.P.); martinsanda@emory.edu (M.G.S.); 4Emory College of Arts and Sciences, Atlanta, GA 30322, USA; 5Department of Radiology and Imaging Sciences, Emory University, Atlanta, GA 30322, USAoabiod2@emory.edu (O.A.A.-O.); dschust@emory.edu (D.M.S.); 6Department of Hematology and Medical Oncology, Emory University, Atlanta, GA 30322, USA; 7Department of Biostatistics and Bioinformatics, Emory University, Atlanta, GA 30322, USA; yhuang5@emory.edu

**Keywords:** cancer genomics, metastasis, prostate cancer, tumor heterogeneity

## Abstract

**Simple Summary:**

We find that mutations in the *TP53*, *FLT4*, *EYA1*, *NCOR2*, *CSMD3*, and *PCDH15* genes are associated with prostate cancer metastasis and mutations in *EYA1* and *CSMD3* are associated with poor outcome. Our finding that oxidative phosphorylation is associated with metastasis and identification of mutations enriched in prostate cancer metastases have implications for our understanding of the molecular events required for prostate metastasis. They also have relevance for our understanding of prostate cancer health disparities and provide a rationale for identification of compounds that target oxidative phosphorylation in metastatic prostate cancer.

**Abstract:**

Prostate cancer is a highly heterogeneous disease and mortality is mainly due to metastases but the initial steps of metastasis have not been well characterized. We have performed integrative whole exome sequencing and transcriptome analysis of primary prostate tumor foci and corresponding lymph node metastases (LNM) from 43 patients enrolled in clinical trial. We present evidence that, while there are some cases of clonally independent primary tumor foci, 87% of primary tumor foci and metastases are descended from a common ancestor. We demonstrate that genes related to oxidative phosphorylation are upregulated in LNM and in African-American patients relative to White patients. We further show that mutations in *TP53*, *FLT4*, *EYA1*, *NCOR2*, *CSMD3*, and *PCDH15* are enriched in prostate cancer metastases. These findings were validated in a meta-analysis of 3929 primary tumors and 2721 metastases and reveal a pattern of molecular alterations underlying the pathology of metastatic prostate cancer. We show that LNM contain multiple subclones that are already present in primary tumor foci. We observed enrichment of mutations in several genes including understudied genes such as *EYA1*, *CSMD3*, *FLT4*, *NCOR2*, and *PCDH15* and found that mutations in *EYA1* and *CSMD3* are associated with a poor outcome in prostate cancer.

## 1. Introduction

Although there have been genomic profiling analyses of primary prostate cancers by The Cancer Genome Atlas [1] and other genomic studies of metastatic castration-resistant prostate cancers (mCRPC) [2,3,4,5,6], analyses of primary tumors and metastases from the same patients are relatively understudied. Most studies have used only primary tumors [1], only metastatic tumors [4,6], or metastases and primary tumors from different patients [3] but only very few have analyzed metastases and primary tumors from the same patients [7]. However, this study used targeted DNA sequencing and low-pass whole genome sequencing for copy number alterations but did not perform whole exome sequencing (WES) or RNAseq analyses [7]. No studies have performed integrative WES and RNAseq using multiple sites from the same patients. Additionally, few studies have analyzed lymph node metastases (LNM) [8] since most studies have focused on bone or visceral metastases [6]. Analysis and understanding of LNM is critical because LNM represents one of the first steps in the systemwide metastatic process and patients with positive lymph nodes at radical prostatectomy experience poor cancer specific survival rates [9]. Furthermore, analyses of multiple independent tumor foci from the same patients together with corresponding LNM are lacking. Moreover, while studies have examined African-American (AA) men at the RNA [10] or DNA levels [11,12], integrative genomic profiling of RNA and DNA has not been performed in highly diverse cohorts with high proportions of AA men. In this study, we address this gap in our knowledge of prostate cancer biology through integrative RNAseq and WES approaches to identify potential key driver mutations and important genomic expression changes involved in prostate cancer metastasis.

Prostate cancer is characterized by intratumoral heterogeneity on multiple levels [13]. Primary prostate cancer is known to be multifocal, which underlies the rationale for the Gleason scoring system [14,15,16,17]. Previous studies have found that over 70% of patients have multifocal disease representing multiple tumor grades [15]. Furthermore, different primary tumor foci can be composed of genetically distinct clones, suggesting independent carcinogenesis events within the prostate gland [18,19,20,21,22]. Intratumoral heterogeneity has been identified even within single primary tumor foci in the prostate [23], suggesting parallel evolution of independent clones with branching and divergent paths.

Intratumoral heterogeneity is a poorly understood phenomenon that is critically important for understanding tumor progression and the development of drug resistance as different sub-clones can respond differently to microenvironmental changes and selection pressure from therapies. Recent studies have indicated that independent foci and biopsies can have markedly different performances in commercially available biomarker panels [24]. Thus, understanding intratumoral heterogeneity is essential for biomarker development and validation, prognosis, and therapeutic decision making for precision medicine.

Here, we have performed WES of 137 samples and RNAseq analysis of 165 samples from 43 patients enrolled in a clinical trial [25] that included both radical prostatectomy and extensive dissection of all pelvic lymph nodes (LNs). We present evidence that while there are some cases of clonally independent primary tumor foci, 87% of primary tumor foci are descended from a common ancestor, and that, while metastases are heterogeneous and contain multiple subclones, they are genetically related to 87% of primary tumor foci within the same patients. We demonstrate that genes related to oxidative phosphorylation are transcriptionally upregulated in LNM and that these genes are also upregulated in African-American patients relative to White patients. We further show that mutations in *TP53*, *FLT4*, *EYA1*, *CSMD3*, and *PCDH15* are enriched in prostate cancer metastases. Moreover, we validate our findings in a meta-analysis of publicly available datasets from twelve different studies representing 3929 primary tumors and 2721 metastases. These findings have implications for our understanding of the molecular events required for prostate metastasis.

## 2. Materials and Methods

### 2.1. Patient Samples

All patient samples were derived from a clinical trial (NCT01808222) conducted at our institution to determine the sensitivity and specificity of [18F]-fluciclovine PET imaging for the detection of significant occult metastatic disease in patients with unfavorable intermediate-risk or high-risk prostate cancer [25]. These patients had no definitive findings of systemic metastases on conventional imaging such as CT, MR, and bone scan. Patients underwent radical prostatectomies and extended pelvic lymph node dissections. In this trial, 56 prostate cancer patients were enrolled based on criteria that correspond to a 50–80% PSA failure rate within high or very high-risk groups (T3a, Gleason score 8–10, or PSA greater than 20 ng/mL) and underwent radical prostatectomy and extended pelvic lymph node dissection. Of these, 30/56 (54%) of patients had metastases. Of those, 7 had only one positive lymph node and 23 had two or more positive lymph nodes. A total of 92 positive LNs were identified out of a total of 2480 excised LNs of which 58 positive LNs were >4 mm in diameter.

Formalin-fixed paraffin-embedded (FFPE) tissue samples were identified from this clinical trial, consisting of prostate and lymph node dissections from 56 cases. Of these 56 cases, 7 did not give consent for genomic analyses (4 metastatic and 3 non-metastatic) and were excluded from the study, leaving a potential of 49 available for genomic studies. For the patients in which the cancer did not metastasize (*n* = 23), we identified at least 3 FFPE prostate tissue samples and 1 LN (benign) for examination. For the cases in which the cancer did metastasize (*n* = 26), we identified at least 4 primary prostate samples, 1 benign LN, and 1 metastatic LN. All available FFPE blocks were collected from Emory University Hospital (EUH) pathology services and coded to remove any personal health information (PHI). For three of the cases, the metastatic LN blocks were not available and were excluded from the study, and for three other cases, no LN metastases exceeded the minimum size of 4 mm and were also excluded, leaving a total of 43 cases available for RNA and DNA extraction. Clinical metadata is provided in Appendix A. A flow chart of inclusion and exclusion of patients is provided in Appendix A. Race was determined by self-identification.

### 2.2. RNA and DNA Extraction and Sequencing

Seven 5 µm sections were prepared at the Winship Cancer Tissue and Pathology Shared Resource Core (CTPSR) for sectioning. One section from each block was stained with hematoxylin and eosin (H&E) for histological analysis by a GU pathologist to identify and annotate prostate cancer regions of interest (ROI) within the tissue samples for macrodissection. Annotated H&E slides were aligned to unstained sections and tissue was collected only from the ROI, which was typically >90% tumor. Entire LNM foci were annotated for macrodissection, excluding any benign regions of the lymph nodes. For separate primary tumor foci, independent foci were identified from contralateral aspects of the prostate or separate locations (e.g., apex vs. base or anterior vs. posterior). The identified regions from all six sections were macrodissected and the tissue was collected directly into a shearing microtube. Once tissue was obtained, RNA and DNA isolation was performed at the Emory Integrated Genomics Core (EIGC) using the Covaris truXTRAC FFPE total NA kit.

Quality control (QC) analysis was also performed by the EIGC on all RNA and DNA extractions and RNAseq and WES were performed by HudsonAlpha (subunit of Discovery Life Sciences). For RNAseq, we performed 100 bp paired-end sequencing using the SMART-Seq Stranded library prep kit (Takara, 634444). For WES analysis, libraries were prepared using the KAPA HyperPrep/Covaris Shearing Library Construction, IDT v2 Capture kit, and 100 bp PE sequencing was performed on a NovaSeq 6000 (Illumina, San Diego, CA, USA). Of the samples prepared, 43 patients had RNA of sufficient quality for sequencing and 35 patients had DNA of sufficient quality for WES. For non-metastatic patients, two primary tumor foci (primary tumor foci) and one benign LN were sequenced. For metastatic patients, three primary tumor foci, one benign LN, and all metastatic LNs were sequenced. RNA was extracted from 10 adjacent normal prostate samples from both metastatic and non-metastatic patients. A total of 165 tissue samples (51 primary tumor foci from metastatic patients, 46 primary tumor foci from non-metastatic patients, 19 LN metastases, 39 benign LNs, and 10 normal adjacent prostate samples) were sequenced via RNA-seq and 137 tissue samples were sequenced via WES (Table 1).

### 2.3. RNA-seq Analysis Pipeline

Raw FASTQ files were analyzed by FastQC for quality control analysis and TrimGalore [26] was used to remove adapter sequences and poor-quality reads (Phred < 24) (31). Next, trimmed reads were mapped with a STAR aligner [27] based on the GRCh38 reference and GENCODE.v.24 annotations. DESeq2 [28] was used to determine differences in RNA expression between sample groups. WebGestalt (WEB-based Gene SeT AnaLysis Toolkit) [29] was used to conduct several gene set enrichment analyses (GSEA) [30] for each comparison. Ecotyper analysis was performed using the Ecotyper website (https://ecotyper.stanford.edu/ accessed on 28 April 2023).

### 2.4. WES Analysis Pipeline

Similar to the RNA-seq pipeline, FastQC was used for quality control checks of the high throughput data and TrimGalore was used to remove adapter sequences and eliminate poor quality reads. Genome mapping was conducted with the Burrows-Wheeler Alignment (BWA) algorithm from the Genome Analysis Toolkit (GATK v4.3.0.0) [31] against the GRCh38 human genome to generate BAM files. GATK and the Picard Toolkit were used to calibrate read group qualities with the command ‘AddOrReplaceReadGroups -LB lib1 -PL illumina’ and mark duplicates with the command ‘MarkDuplicatesSpark’ (39). The GATK Mutect2 algorithm was used to call somatic SNVs and indels for all tumor samples for a patient relative to the benign lymph node sample from that patient. Additionally, a panel of normals (PON) [1] composed of all benign lymph node samples was used in the Mutect2 command and a minimal callable depth of 40 was used. Somatic mutations were further filtered with the gatk command ‘FilterMutectCalls’. VCF files were further analyzed using custom scripts in R to identify significant mutations based on a minimum read depth of 40 reads per base, an allele frequency (AF) difference > 0.05, and false discovery rate (FDR) < 0.1 based on a Fisher Exact Test used to determine if the allele frequency in the tumor sample was different than the normal sample. GATK funcotator was used to annotate mutations and generate mutation annotation format (MAF) files. These MAF files were analyzed using the R package maftools [32] to identify recurrent mutations and patterns to build mutational profiles. BAM files were also analyzed using HATCHet [33], an algorithm to compute tumor heterogeneity, copy number aberrations (CNAs), and whole-genome duplications (WGDs) used to investigate tumor evolution and metastatic seeding patterns. Hatchet version 1.2.1 was installed and the complete WES pipeline scripts were used with GATK BAM files as the input using default settings with the following modifications. Autosomal chromosomes (chr1-chr22) were analyzed using default parameters except the following: mincov = 10, maxcov = 1000, msr = 200, mtr = 200, and blocklength = 50 kb. All steps (download_panel, count_reads, genotype_snps, count_alleles, combine_counts, cluster_bins, loc_clust, plot_bins, compute_cn, and plot_cn) were set to TRUE.

## 3. Results

To investigate the molecular changes, mutations, and signaling pathways that characterize the initial steps of prostate cancer metastasis to the lymph nodes, we analyzed 19 discrete LNM, 97 primary tumor foci, 39 benign lymph nodes, and 10 normal adjacent prostate tissue samples from 43 patients enrolled in a clinical trial (NCT01808222) of extended lymphadenectomy and metastasis excision guided by [18F]-fluciclovine PET imaging for high risk newly diagnosed prostate cancer conducted at our institution [25] (Appendix A). These patients fell within high or very high-risk groups (T3a, Gleason score 8–10, or PSA greater than 20 ng/mL). Of these patients, 16 had LN metastases at least 4 mm in diameter and 20 patients had no metastases. For metastatic patients, we performed RNA and whole exome sequencing (WES) on three primary tumor foci, one benign LN, and all LN metastases greater than 4 mm in diameter (see Section 2). For non-metastatic patients, we sequenced two primary tumor foci and one benign LN. In total, RNAseq was performed on 165 tissue samples from 43 patients (Appendix A) and WES was performed on 137 samples from 35 patients (Appendix A). A summary of all sequenced samples is provided in Table 1.

### 3.1. RNA-seq Analysis Identifies Oxidative Phosphorylation Is Associated with LNM

To understand the signaling pathways essential for the first steps of metastasis, we performed RNAseq analysis. A median of 56 million paired-end reads were obtained per sample and 44,289 transcripts were detected in at least 5% of samples. Samples were classified by several criteria including the tissue site of origin (prostate vs. lymph node), malignant vs. benign, primary tumor vs. metastasis, and deriving from a patient with metastases vs. a patient without metastases (met vs. non-met).

In the first analysis, we identified a set of 381 genes whose expression trended consistently upward, significantly increasing from normal prostate tissue to primary tumor foci and from primary tumor foci to LNM as well as a set of 323 genes that trended consistently downward from normal tissue to primary tumor foci to LNM (Figure 1A and Appendix A). The upregulated gene set included oncogenes such as PIK3CB, FGFRL1, and NCOA2; prostate cancer-related genes such as SPON2, PCAT4, and SCHLAP1; and cell cycle genes such as MKI67, MCM4, KIF4A, CENPF, and FLNB. Downregulated genes included tumor suppressors such as PTEN and TGFBR3; adhesion and junction proteins such as SDC3, ITGA9, ITGA7, RSU1, and TGFBI; and apoptotic genes such as TNFRSF10B (DR5/TRAILR2) and RAPGEF3 (EPAC1). These consistent changes in gene expression across sample types suggest that they play key roles in metastasis to the lymph nodes.

Next, we compared 39 benign lymph nodes (BLN) to 19 LNM and identified 9396 transcripts differentially expressed (*p*-adj < 0.01) between these two sets of samples (Appendix A). As expected, the most differentially expressed genes included NKX3-1, KLK3 (PSA), KLK2, KLK4, TMPRSS2, AR, AMACR, FOLH1 (PSMA), and PCA3. Also of interest were several genes associated with prostate cancer metastases including FOXA1, EZH2, SCHLAP1, CDH1, SOX4, SOX9, HOXB13, and TGFBR3. Gene set enrichment analysis (GSEA) identified several pathways significantly associated with LNM including the Hallmark 50 pathways ANDROGEN_RESPONSE, MYC_TARGETS_V1, OXIDATIVE_PHOSPHORYLATION, MTORC1_SIGNALING, and MYC_TARGETS_V2, which all had an FDR < 2.2 ×10^−16^ (Table 2 and Figure 1B). Many of the pathways listed in Table 2 have been extensively studied and validated in the context of prostate cancer, lending confidence in the quality of our data. However, oxidative phosphorylation has been less studied in the context of prostate cancer metastasis and this pathway was found to be significantly altered in multiple additional analyses. Altered expression of the oxidative phosphorylation pathway was also observed to be differentially expressed in AA men and in metastatic clones (see below). 

Comparing primary tumor foci to LNM, 6203 transcripts were differentially expressed (*p*-adj < 0.01) (Appendix A). Primary tumor foci were enriched relative to LNM in gene sets associated with the epithelial to mesenchymal transition (EMT), estrogen signaling, TGF beta signaling, Notch signaling, and the response to hypoxia. LNM were enriched in cell cycle progression pathways and immune function pathways such as interferon (IFN) gamma and IFN alpha responses (Appendix A).

### 3.2. Oxidative Phosphorylation, Immune Signaling, and Hedgehog Pathways Are Differentially Expressed in AA Patients

The racial diversity of our cohort enabled us to identify differences in gene expression between (self-identified) African-American and White patients. Of the 165 samples analyzed by RNAseq, 30 primary tumor foci samples came from 14 African-American patients and 53 primary tumor foci samples came from 23 White patients. There were only three LNM from three African-American patients and 14 LNM from 12 White patients. To optimize sample power, we chose to focus on primary tumor foci to examine differences in gene expression based on race. We determined that the average Gleason score from the radical prostatectomies was higher in the White patients than in the African-American patients (8.61 vs. 7.93, *p* = 0.03) and. therefore, when computing differences in gene expression between primary tumor foci from African-American and White patients, we performed subset analyses on patients with a Gleason score of 7 (16 samples from 7 African-American patients and 7 samples from 4 White patients) and a Gleason score of 9 (12 samples from 6 African-American patients and 43 samples from 18 White patients). We performed GSEA analysis on these differences and determined that for Gleason score 7 patients, primary tumor foci from African-Americans were enriched in gene sets relating to the IFN alpha response, oxidative phosphorylation, and adipocyte development, among others (Figure 1C, Appendix A). For Gleason score 9 patients, primary tumor foci from African-Americans were enriched in gene sets relating to Hedgehog signaling, muscle differentiation, and EMT (Appendix A). For both sets of analyses, primary tumor foci from African-Americans were enriched in gene sets associated with the UV response, early estrogen response, adherens and tight junctions, and mitotic spindle assembly (Appendix A). Differential expression of the DNA damage response, hormonal response, adhesion, and proliferation pathways in AA men are consistent with the typically worse outcomes observed in these patients [34]. In addition, we performed GSEA on all primary tumor foci from African-American and White patients controlling for Gleason score (Appendix A) and found that oxidative phosphorylation was highly significant (FDR < 2.2 × 10^−16^) along with the interferon alpha response, UV response, adipogenesis, mitotic spindle, and estrogen response.

Three of the most significantly upregulated OXPHOS genes include IDH1, IDH2, and FGFR3. These data are intriguing in light of a recent study on the role of the FGFR3-TACC3 gene fusion driver mutation that has been observed in 3% of human glioblastoma cases [35]. This study found that the FGFR3-TACC3 fusion protein drove increased mitochondria, ATP production, and oxidative phosphorylation [35]. To better quantify the OXPHOS pathway activation in our RNAseq datasets, we constructed an OXPHOS score based on the z-score normalized gene expression values of the 105 genes in the leading edge of the OXPHOS gene set analyses. We computed the OXPHOS score for all samples and compared the expression in normal, primary tumor, and LNM samples and found increased OXPHOS scores as tumors progressed (Figure 1D). Interestingly, FGFR3 was highly correlated with OXPHOS gene expression, including expression of IDH1 (Figure 1E, *p* < 2.2 × 10^−16^) and IDH2 (*p* = 3.7 × 10^−14^), which is critical for and modulates the assembly of the entire OXPHOS system [36]. We noted that FGFR3 was expressed at higher levels in AA samples than in EA samples in our cohort (Figure 1F, *p* = 0.01, fold change = 1.82). To investigate whether this observation was confirmed in an independent cohort, we also analyzed the association of FGFR3 mRNA expression with %African Ancestry data from the cancer genome atlas (TCGA) prostate adenocarcinoma (PRAD) dataset [1] using data available on cBioPortal [37,38] and determined that FGFR3 was one of the genes most strongly associated with %African (%AFR) ancestry (Spearman rho = 0.2, *p* = 1.479 × 10^−5^). We also determined that FGFR3 expression was higher in patients with >50% AFR ancestry compared to patients with <50% AFR ancestry (Figure 1G, adj.*p* = 0.027).

### 3.3. Ecotyper Analysis Identifies That Ecotypes with Activated Oxidative Phosphorylation and Androgen Signaling Are Associated with LNM

The Ecotyper tool [39] is a machine learning framework for analysis of gene expression patterns from bulk tumor RNA to identify cellular subtypes and the state of the tumor microenvironment. We applied Ecotyper analysis to our RNAseq expression data using the 10 carcinoma ecotypes (CE) that were generated using 5946 tumors from 16 TCGA studies [39]. The Ecotyper tool was able to assign a dominant carcinoma ecotype to 131 of the 165 RNAseq samples (Figure 1H and Appendix A). In general, the CEs that were assigned to each sample were consistent with the tissues from which they were derived. For example, samples from the lymph node were primarily assigned to CE10, which is associated with IL-2, IL-6, and allograft rejection, and sometimes CE9, which is associated with IFN-alpha and IFN-gamma signaling. Additionally, normal prostate tissue samples were most often assigned to CE6, which is associated with normal tissues. LNM was also assigned to CE5, CE7, and CE8. CE5 is associated with spermatogenesis, MYC signaling, and DNA repair pathways, while CE7 is associated with Androgen, PI3K, MTORC, and OXPHOS pathways and CE8 is associated with spermatogenesis and CCL16. Many of the primary tumor foci were associated with CE1 and CE2, which are associated with EMT, TGFB, Notch, and Hedgehog signaling in CE1 and hypoxia and proliferation in CE2. CE7 and CE8 were enriched in patients who had metastases. For CE7 samples, 73% of samples were from patients with metastases, while for CE8, 67% of samples were from patients with metastases. The Association of CE7 with metastatic samples is consistent with previous studies demonstrating the activation of androgen and PI3K/MTOR pathways in mCRPC [3,9] and our identification here of the oxidative phosphorylation pathway association with LNM.

### 3.4. WES Analysis Determines That SPOP, FLT4, and PCDH15 Mutations Are Enriched in LNM

To identify mutations associated with LNM, we performed a WES analysis. A total of 137 tissue samples from 35 patients were subjected to WES analysis at a median exome depth of 149 reads. GATK analysis pipelines including BWA alignment and Mutect2 variant calling using a panel of 34 normal samples were used to generate variant calls as previously described [1] (see Section 2). We identified 1927 significant somatic mutations in 91 patient samples (Appendix A). The top mutated genes based on the number of samples across each site are shown (Figure 2A). The top 10 mutated genes included SPOP, EYA1, NCOR2, CACNA1E, SOGA1, CSMD3, TP53, PCDH15, LRRC4C, and FOXP1 (Figure 2B). We observed four different missense mutations in SPOP from five patients, all of which were located in the MATH domain, which is involved in receptor binding and oligomerization (Figure 2C). Two patients (PCM021 and PCM033) had the same SPOP-F133V mutation. Furthermore, we identified specific mutations that were enriched in metastatic samples using maftools and determined that genes preferentially mutated in metastases included SOGA1, LRRC4C, TP53, COL5A1, PCDHA13, and SLC16A14 (Figure 2D). Mutations were most frequent in RTK/RAS, Wnt, PI3K, Hippo, and Notch pathways (Figure 2E).

To examine tumor heterogeneity, we performed clustering based on mutation allele frequencies. Mutational allele frequency clustering identified some cases of clonally independent primary tumor foci. While some patients (e.g., PCM034 and PCM003) had primary tumor foci that appear to be clonally independent from the metastases and other primary tumor foci (Figure 3A), most primary tumor foci (e.g., PCM037 and PCM050) were clonally related to the metastases and derived from a common ancestral clone (Figure 3A). We also identified mutations that were significantly different in allele frequency between LNM and all primary tumor foci from the same patients. We found that mutations in FLT4, LRRC4C, PCDH15, FAM47B, PIK3CD, MAP3K15, PIK3R6, and SPOP, among others, were significantly enriched in LNM relative to primary tumor foci, suggesting natural selection for these somatic mutations in metastatic clones (Figure 3B). Of those genes, FLT4, SPOP, and PCDH15 were mutated in multiple patients. The association of FLT4 and PCDH15 with prostate cancer metastases has not been previously reported.

### 3.5. Integrative DNA/RNA Analysis Determines the Oxidative Phosphorylation Pathway, FLT4 Mutations, and PCDH15 Mutations Are Associated with Metastasis

To better understand molecular changes associated with metastasis, we performed integrated WES and RNAseq analyses. Based on shared genetic alterations of primary tumor foci to LNM, we categorized samples as either metastatic clones or non-metastatic clones (Appendix A) by examining the number of shared mutations and estimating if they were genetically related as being descended from a common ancestor containing the same somatic mutations. We performed DESeq2 analysis comparing these two groups to control for the fact that primary tumor foci and LNM are derived from different tissue microenvironments and different organ sites. We compared primary tumor foci from metastatic clones (primary tumor foci.met) to primary tumor foci from non-metastatic clones (primary tumor foci.nonmet) and both sets of primary tumor foci to normal prostate tissue (NP). Primary tumor foci.met samples were categorized based on mutation allele frequency as compared to LNM to determine if they were clonally related. Primary tumor foci.nonmet samples were either derived from patients without metastases or from patients with metastases that were not clonally related to corresponding LNM. The number of differentially expressed genes in each comparison is shown in the Venn Diagram in Figure 3C. Over-representation analysis of the 810 genes at the intersection of the three comparisons (i.e., genes different between all three groups of normal prostate and non-metastatic primary tumor and metastatic primary tumor samples, Appendix A) determined that they are enriched in genes involved in oxidative phosphorylation (Figure 3D). These genes included multiple ATP synthase genes (ATP5MF, ATP6V0B, and ATP6V1G1), multiple cytochrome C oxidase and reductase genes (COX17, COX6A1, COX6B1, UQCRH, and UQCRQ), and multiple NADH:ubiquinone oxidoreductase subunits (NDUFA3, NDUFB7, NDUFS6, and NDUFS8).

To further uncover gene expression differences associated with metastatic clones, we compared the metastatic primary tumor clones to the set of normal prostate and non-metastatic primary tumor clones. This analysis identified 1962 significantly different genes (*p*.adj < 0.05). GSEA analysis of these genes identified 6 gene sets associated with metabolism, including oxidative phosphorylation (hsa00190, FDR < 2.2 × 10^−16^), 7 gene sets associated DNA repair, 12 gene sets associated with cell cycle, 7 gene sets associated with immune responses, and 3 gene sets associated with protein trafficking (Appendix A). A summary of the multiple strands of evidence supporting oxidative phosphorylation as a critical pathway in prostate cancer metastasis and cancer health disparities is provided in Appendix A.

In addition, we analyzed associations of mutant alleles to Ecotyper carcinoma ecotypes. We restricted our analysis to mutations that were observed in at least two different patients and computed adjusted *p*-values using Bonferroni correction. Mutations in FLT4 were significantly associated with assignment of samples to CE7 (adj-*p* = 1.1 × 10^−4^) and CE8 (adj-*p* = 1.5 × 10^−3^), while mutations in PCDH15 were also significantly associated with the CE7 (adj-*p* = 7.8 × 10^−7^) and CE8 (adj-*p* = 1.7 × 10^−3^) ecotypes that included many LNM samples.

### 3.6. Tumor Heterogeneity Analysis Identifies Subclones Suggesting the Gain of chr8 Prior to chr8p Loss

To examine subclones within each sample, we performed HATCHet analysis. The HATCHet software tool can compute tumor heterogeneity [34], copy number aberrations (CNAs), and whole-genome duplications (WGDs) to investigate tumor evolution and metastatic seeding patterns. We applied HATCHet analysis to our WES dataset and observed that most samples had subclonal populations (Figure 4). Patient PCM034, which had three LNM, was of particular interest and had three subclones. Clone 1 was enriched in all three LNM and was characterized by a gain of chromosomal arm 8q (containing c-MYC) and loss of chromosomal arm 8p (containing NKX3.1). The other subclones exhibited a gain of chr8 but no loss of 8p, suggesting that the gain of chr8 may have preceded the loss of 8p during tumor evolution in this patient.

### 3.7. Validation in External Datasets Demonstrates That EYA1 and CSMD3 Mutations Are Associated with Poor Survival

To evaluate the generalizability of our findings, we analyzed the mutations that we identified that appeared to be more frequent in LNM than in primary tumor foci or were frequently mutated in our cohort. We examined data from 12 published studies [2,3,6,40,41,42,43,44,45,46,47,48] that are publicly available in cBioPortal [36,37], comprising 8902 samples. Of those 8902 samples, 6650 samples (Appendix A) were definitively annotated as either primary tumors (*n* = 3929) or metastases (*n* = 2721). We found that mutations were enriched in metastases over primary tumors for TP53 (q = 2 × 10^−64^), EYA1 (q = 2 × 10^−9^), CSMD3 (q = 6.1 × 10^−6^), MAP3K15 (q = 3.3 × 10^−5^), PIK3CD (q = 2.2 × 10^−4^), SLC16A14 (q = 1.4 × 10^−3^), FLT4 (q = 1.6 × 10^−3^), PCDH15 (q = 2.1 × 10^−3^), CACNA1E (q = 1.5 × 10^−2^), and NCOR2 (q = 1.9 × 10^−2^) by a Fisher exact test adjusted for false discovery (Figure 5A). We found SPOP mutations were enriched in primary tumors (q = 3.1 × 10^−3^) and, surprisingly, LRRC4C were also enriched in primary tumors (q = 3.1 × 10^−5^). The complete list of mutations from this meta-analysis enriched in metastases or primary tumors includes well-established oncogenes and tumor suppressors such as AR, TP53, PTEN, RB1, APC, MYC, and CTNNB1 and is provided in Appendix A. Mutational comparisons for EYA1 and CSMD3 are shown in Figure 5B,C. To examine the clinical significance of these mutations, we compared the disease-specific survival (492 patients) in the TCGA PanCancer dataset [45], disease-free survival (334 patients), and progression-free survival (494 patients) for samples with and without EYA1 (Figure 5D–F) or CSMD3 (Figure 5G–I) mutations (including SNV, CNV, and structural variations) and found that mutations in both these genes were significantly associated with poor outcomes. FLT4 had only five mutated samples in the TCGA PRAD dataset and, thus, did not have enough events to provide statistically meaningful data and was not statistically significant. PCDH15 had more mutated samples (*n* = 18) but did not indicate statistical significance for disease-free survival, progression-free survival, overall survival, or disease-specific survival.

## 4. Discussion

Mortality from prostate cancer is due to metastases and, thus, understanding the mechanisms of metastasis is essential for improving patient outcomes. The heterogeneity of metastases is poorly understood and there are multiple opposing models of prostate cancer metastasis. Some data using copy number analyses support a monoclonal model of metastasis in which most metastatic lesions derive from a single clone despite the multiclonal nature of the primary tumor [49]. Additional studies have shown limited genomic diversity in multiple metastases from the same men [50]. However, conflicting studies support polyclonal seeding of metastases based on the whole genome sequencing of 51 tumors from 10 patients [51]. Some of the differences between these studies may be due to varying degrees of detail and granularity in the methods employed to molecularly characterize prostate cancer metastases. In addition, it is not clear if the genomic variability observed in metastases in some studies is due to mutations that occur at the metastatic site or if polyclonal populations from different primary foci seeded those metastases. Most studies that have analyzed intratumoral heterogeneity have examined a limited number of patients and used only the index lesion of the primary tumor.

Here, we aimed to address this gap in our understanding of heterogeneity in prostate cancer metastasis and to leverage the unique resources of many patients from a clinical trial that includes both radical prostatectomy and extensive dissection of all pelvic LNs. Since metastasis to surrounding LNs is one of the first steps in metastatic spread, understanding intratumoral heterogeneity in pelvic LNs provides unique insights into the initial mechanisms and heterogeneity of prostate cancer metastasis. Additionally, analysis of multiple primary foci greatly increases the richness of our dataset and enables an integrated functional and clinical genomics approach to reveal genes driving aggressive metastatic prostate cancer.

Our data suggest that LNM contains multiple subclones that are already present in multiple related primary tumor foci. However, some subclones are more abundant in the LNM than in primary tumor foci, suggesting natural selection for these subclones and enrichment for specific mutations. We also show, here, that in addition to well-established oncogenes such as *AR* and *TP53*, mutations in lesser studied genes such as *EYA1*, *CSMD3*, *FLT4*, *NCOR2*, and *PCDH15* are enriched in prostate cancer metastases relative to primary tumors. Additionally, mutations in *FLT4* and *PCDH15* are associated with the CE7 and CE8 gene expression ecotypes associated with Androgen, PI3K, MTORC, and OXPHOS pathways. Finally, using publicly available datasets, we have shown that mutations in *EYA1* and *CSMD3* are not only more frequent in metastases but are also associated with worse disease-specific survival, progression-free survival, and disease-free survival.

The most frequent somatic mutation that we observed occurred in the Speckle-type POZ protein (*SPOP*) gene, which is essential for ubiquitination and subsequent proteasomal degradation [52,53]. Functional SPOP is necessary for the degradation of AR and mutated SPOP fails to ubiquitinate AR, allowing for an increase in AR protein [52,53]. Consistent with previous findings, we observed recurrent mutations in the MATH binding domain of *SPOP*, which mediates protein–protein interactions with AR. Recent studies have also identified important interactions between *SPOP* and c-JUN [54], in which mutated SPOP can bind to and stabilize c-JUN, potentially leading to accelerated cell proliferation. Nevertheless, SPOP mutations are more frequent in primary tumors than metastases and patients with SPOP mutations have improved outcomes [2].

Our findings are consistent with previous studies and expand upon them further. While we did not observe driver mutations in KIF4A and WDR62 that have been observed in metastatic prostate cancer [55], we did observe mutations in KIF5C, KIF7, KIF14, KIF18A, KIF20B, and KIF22 as well as in WDR17, WDR78, and WDR87, suggesting that other members of these families may also serve as cancer drivers. Consistent with earlier studies, we also observed mutations in FOXA1 [2,3,5], NCOR2 [9], and APC [9].

The nuclear receptor corepressor 2 (*NCOR2*) gene encodes a nuclear co-repressor (NCOR2) that mediates gene silencing. NCOR2 interacts with nuclear receptors, such as AR, to promote gene repression. Recently, it has been shown that reduced NCOR2 expression accelerates failure of androgen deprivation therapy in prostate cancer patients [56]. Mutations in *NCOR2* could interfere with the ability of NCOR2 to regulate and maintain the epigenome and result in worse patient outcomes.

EYA transcriptional coactivator and phosphatase 1 (*EYA1*) encodes a transcription factor on chr8 that interacts with the SIX1 transcription factor and has intrinsic phosphatase activity important for SIX1 activity during development [57]. Mutations in *EYA1* may cause dysregulation and act as a tumor promoter with SIX1 via activation of STAT3 signaling in thyroid carcinomas [58].

CUB and Sushi multiple domains 3 (*CSDM3*) have been shown to be one of the 10 most recurrently mutated genes in prostate cancer [59] and SNPs in *CSMD3* are associated with the risk of prostate cancer in Hispanic men [60]. CSMD3 is also located on chr8 and *CSMD3* was identified as the second most frequently somatically mutated gene (next to TP53) in non-small cell lung cancer [61]. *CSMD3* mutation is highly correlated with increased tumor mutational burden and poor clinical prognosis in ovarian cancer [62].

FMS-related receptor tyrosine kinase 4 (*FLT4*, also known as VEGFR-3) is a receptor tyrosine kinase that binds to VEGF-C and VEGF-D. Mutations in *FLT4* have been associated with sentinel lymph node metastases in prostate cancer [63] and FLT4 plays a critical role in prostate lymphangiogenesis [64]. Our observation that FLT4 mutations are associated with carcinoma ecotypes is intriguing and future studies will examine a potential causative role for these mutations.

Protocadherin-related 15 (*PCDH15*) is a member of the cadherin superfamily that mediates cell–cell adhesion. Somatic mutation of PCDH15 has been associated with metastasis in ocular adnexal sebaceous carcinoma [65]. PCDH15 knockdown significantly increases ERK phosphorylation and activation, increasing the proliferation of oligodendrocyte progenitor cells [66]. Our observation that PCDH15 mutations are associated with specific carcinoma ecotypes is an avenue for future investigation.

Through our RNAseq analyses, we determined that genes that were differentially expressed between normal prostate, metastatic clones in the prostate, and non-metastatic clones in the prostate were enriched for oxidative phosphorylation. Interestingly, lower grade African-American samples with a Gleason score of 7 were also enriched for this gene set. Both low and high-grade African-American samples had increased expression of genes associated with adherens, tight junctions, and mitotic spindle assembly, which could be related to the generally more aggressive phenotypes and worse outcomes associated with African-American patients [67,68].

There are a number of limitations to this study. First, while this study includes tissues from only 43 patients, of which only 16 had metastases, it does provide RNAseq analysis of 165 samples and WES analysis of 137 samples, with at least three and up to seven samples for each patient. Second, the samples used were macrodissected from FFPE blocks and, thus, were analyzed by bulk sequencing. Finally, while many of our observations were either consistent with the previous literature or validated in large external datasets, functional validation of the observed mutations and gene expression changes are needed to better understand the mechanisms of metastasis and poor outcome for prostate cancer patients.

## 5. Conclusions and Future Directions

The data provided in this study can serve as a resource for future studies of prostate cancer metastasis and tumor heterogeneity. We show that LNM contains multiple subclones that are already present in multiple related primary tumor foci and suggest natural selection for some metastatic subclones and enrichment for specific mutations. We further show that oxidative phosphorylation is strongly associated with both metastasis and racial disparities. Integrative genomic analysis of primary prostate tumors and associated LNM demonstrated the enrichment of mutations in several genes including both well-established oncogenes and tumor suppressors and understudied genes such as *EYA1*, *CSMD3*, *FLT4*, *NCOR2*, and *PCDH15*. Future studies of these genes could provide novel avenues for the detection of patients at high risk for metastasis as well as new therapeutic approaches specifically targeting the metastatic process. Future directions for this work include further characterization of *EYA1*, *CSMD3*, *FGFR3*, and *PCDH15* in prostate cancer metastases and the roles of oxidative phosphorylation and *FGFR3* in prostate cancer health disparities. We also plan to correlate gene expression patterns with [18F]-fluciclovine uptake.

## Figures and Tables

**Figure 1 cancers-15-05671-f001:**
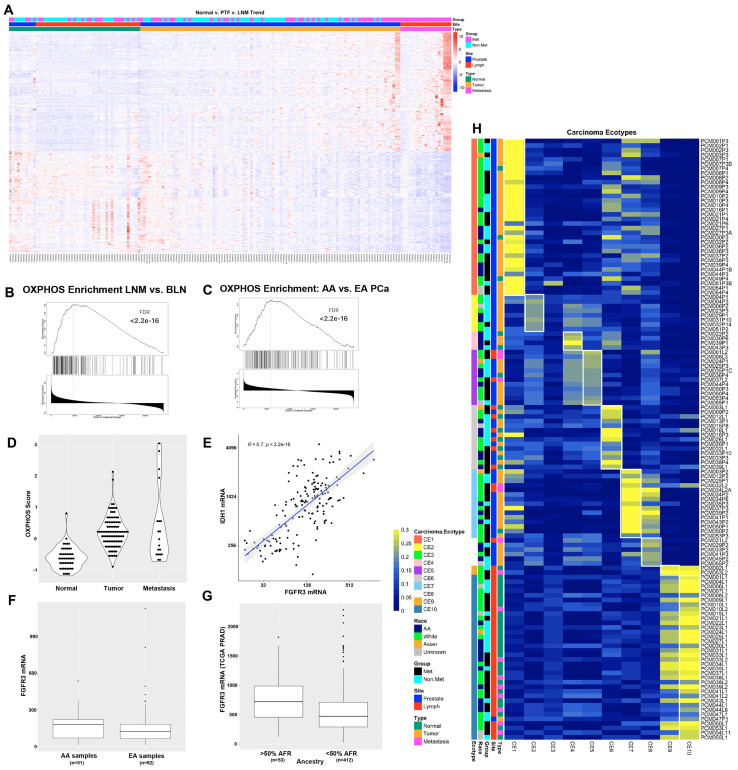
RNAseq analysis of 165 prostate cancer patient samples identifies oxidative phosphorylation as differentially expressed between AA and White patients and between LNM and benign lymph nodes. (**A**) Heat map of 704 genes that trend consistently upward or downward from normal to primary tumor foci to LNM. (**B**) GSEA plot of the oxidative phosphorylation gene set from a comparison of LNM to benign lymph nodes. (**C**) GSEA plot of the oxidative phosphorylation gene set from analysis of African-American vs. White patients with Gleason scores of 7. (**D**) OXPHOS score increases in from normal prostate to primary tumor to LNM. (**E**) FGFR3 is highly correlated with OXPHOS gene IDH1. (**F**) FGFR3 is expressed at higher levels in AA PCa than EA PCa in our cohort. (**G**) FGFR3 mRNA is significantly higher in patients with >50% African ancestry in the TCGA PRAD dataset. (**H**) Ecotyper analysis of 131 samples assigned to one of ten unique carcinoma ecotypes (CE).

**Figure 2 cancers-15-05671-f002:**
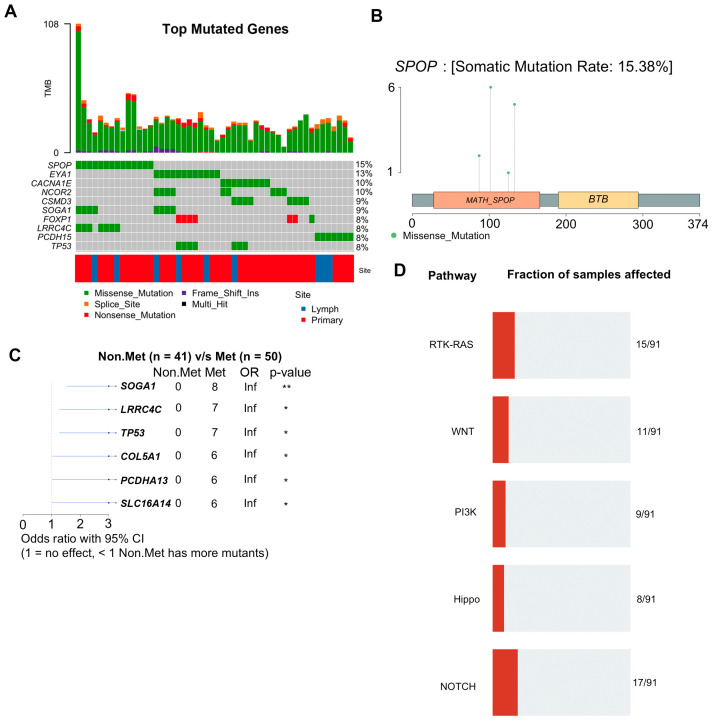
WES analysis of 137 prostate cancer patient samples identifies recurrent mutations associated with metastasis. (**A**) Graphical depiction of the top ten most frequently mutated genes in this study. Each column represents an individual tissue sample. Percentages on the right indicate the percentage of samples with mutations in that gene. Tumor mutational burden (TMB) for each sample is plotted along the top. (**B**) Lollipop plot of mutations detected in the SPOP gene. (**C**) Forest plot of mutations preferentially detected in patients with metastases. * <0.05; ** <0.01 (**D**) Signaling pathways most highly impacted by the detected mutations.

**Figure 3 cancers-15-05671-f003:**
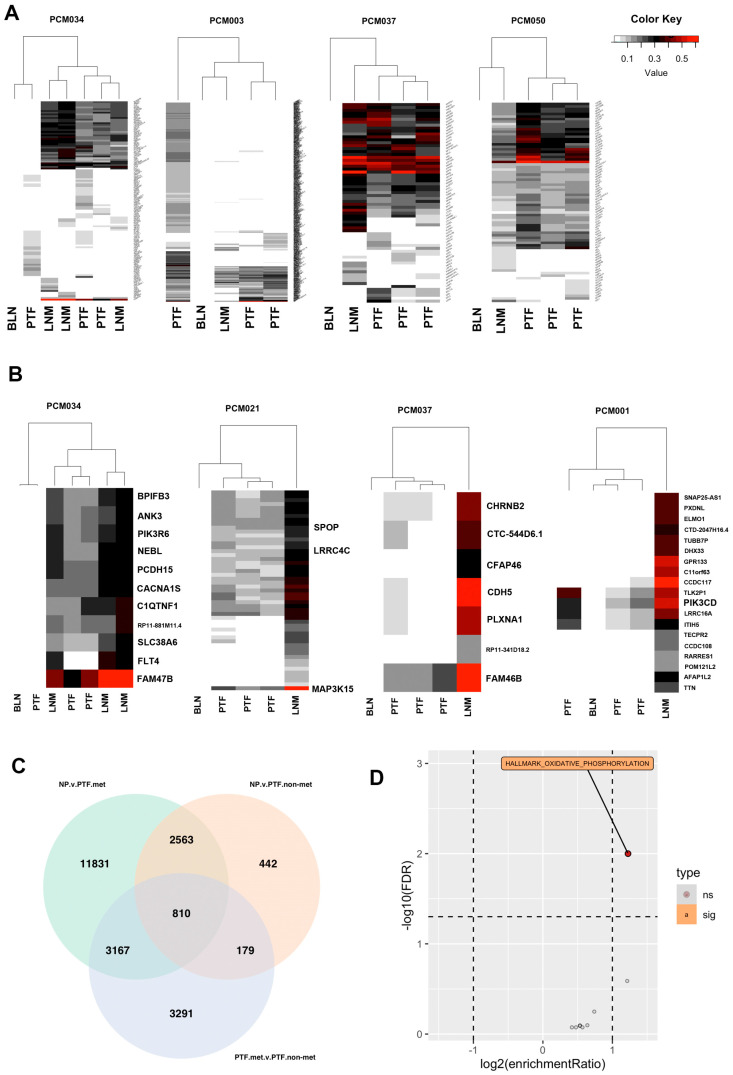
Analysis of metastatic and non-metastatic clones identifies oxidative phosphorylation as associated with metastasis. (**A**) Heatmaps of all identified mutations clustered based on allele frequencies. LNM are clonally related to most primary tumor foci. Some patients (PCM034 and PCM003) contained primary tumor foci that were not clonally related to others, although most patients did not. (**B**) Heatmaps of genes with significantly increased allele frequencies in LNM relative to primary tumor foci within the same patients indicating potential clonal selection. (**C**) Venn diagram of sets of significantly differentially expressed genes comparing normal prostate to metastatic primary tumor foci, normal prostate to non-metastatic primary tumor foci, and metastatic primary tumor foci to non-metastatic primary tumor foci. (**D**) Volcano plot of over-representation analysis of the 810 genes in the center of the Venn Diagram in panel C. Oxidative phosphorylation was the only significantly overrepresented pathway.

**Figure 4 cancers-15-05671-f004:**
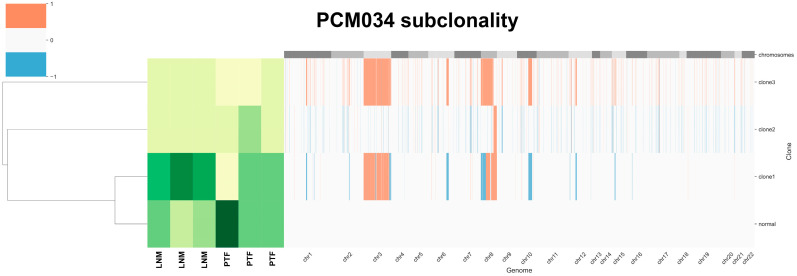
HATCHet analysis of tumor heterogeneity for patient PCM034 identifies subclones with differential gains in chr8. Gains and losses for each chromosome are indicated as red or blue. The abundance of subclones is indicated with lighter colors showing less abundance and darker colors showing greater abundance. Four subclones were identified, with clone 1 containing a loss of chr8p and gain of chr8q being most prevalent in the LNM samples. Primary tumor foci were enriched in the normal subclone.

**Figure 5 cancers-15-05671-f005:**
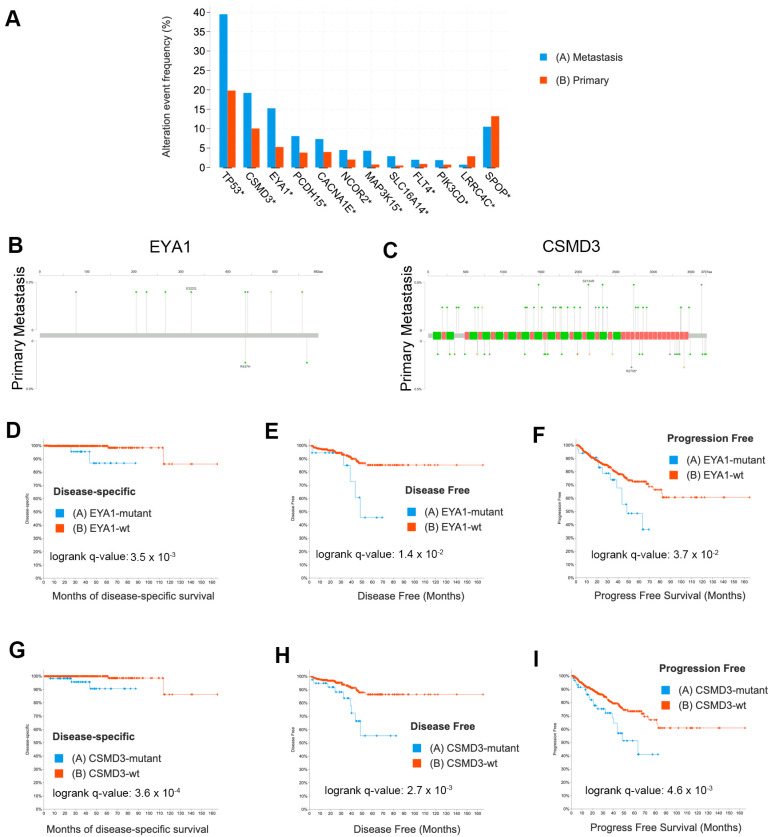
Validation of mutations using external datasets analyzed in cBioPortal demonstrates EYA1 and CSMD3 mutations are associated with patient outcome. (**A**) Enrichment of mutations in metastases (*n* = 2721) or primary tumors (*n* = 3929). Asterisk (*) indicates significant q-value based on Fisher exact tests. (**B**) Lollipop plot of EYA1 mutations in metastases (top) compared to primary tumors (bottom). (**C**) Lollipop plot of CSMD3 mutations in metastases (top) compared to primary tumors (bottom). (**D**) Kaplan–Meier plot of disease-specific survival comparing EYA1 mutant cases (*n* = 34) to EYA1 wild-type cases (*n* = 458). (**E**) Kaplan–Meier plot of disease-free survival comparing EYA1 mutant TCGA cases (*n* = 18) to EYA1 wild-type TCGA cases (*n* = 316). (**F**) Kaplan–Meier plot of progression-free survival comparing EYA1 mutant TCGA cases (*n* = 34) to EYA1 wild-type TCGA cases (*n* = 460). (**G**) Kaplan–Meier plot of disease-specific survival comparing CSMD3 mutant TCGA cases (*n* = 63) to CSMD3 wild-type TCGA cases (*n* = 429). (**H**) Kaplan–Meier plot of disease-free survival comparing CSMD3 mutant TCGA cases (*n* = 39) to CSMD3 wild-type TCGA cases (*n* = 295). (**I**) Kaplan–Meier plot of progression-free survival comparing CSMD3 mutant TCGA cases (*n* = 63) to CSMD3 wild-type TCGA cases (*n* = 431).

**Table 1 cancers-15-05671-t001:** Sample distribution across 40 prostate cancer patients.

Samples/Patients	RNAseq	WES	Both
Primary Tumor Foci	97	88	80
Metastatic Lymph Nodes (LNM)	19	19	19
Normal Prostate tissue (NP)	10	0	0
Normal Lymph Nodes (LN)	39	37	33
Total Samples	165	137	132
Metastatic Patients	18	15	15
Non-Metastatic Patients	25	20	20
White Patients	23	20	20
African-American Patients	14	13	13
Total Patients	43	35	35

**Table 2 cancers-15-05671-t002:** GSEA Analysis of BLN vs. LNM.

Gene Set	Size	Leading Edge Number	NES	FDR
HALLMARK_ANDROGEN_RESPONSE	101	47	−2.3952	<2.2 × 10^−16^
HALLMARK_MYC_TARGETS_V1	200	132	−2.2692	<2.2 × 10^−16^
HALLMARK_OXIDATIVE_PHOSPHORYLATION	199	117	−2.1852	<2.2 × 10^−16^
HALLMARK_MTORC1_SIGNALING	200	120	−2.0443	<2.2 × 10^−16^
HALLMARK_MYC_TARGETS_V2	58	35	−1.9282	<2.2 × 10^−16^
HALLMARK_FATTY_ACID_METABOLISM	157	70	−1.8465	0.000112
HALLMARK_GLYCOLYSIS	199	93	−1.8362	0.000096004
HALLMARK_UNFOLDED_PROTEIN_RESPONSE	113	55	−1.8126	0.00033601
HALLMARK_PEROXISOME	104	54	−1.806	0.00037335
HALLMARK_E2F_TARGETS	200	103	−1.7584	0.00039202
HALLMARK_EPITHELIAL_MESENCHYMAL_TRANSITION	200	73	1.6463	0.0029856
HALLMARK_MYOGENESIS	197	69	1.7325	0.002488
HALLMARK_KRAS_SIGNALING_UP	198	79	1.7459	0.0023325
HALLMARK_TNFA_SIGNALING_VIA_NFKB	200	88	1.8804	0.00053315
HALLMARK_INFLAMMATORY_RESPONSE	199	86	1.9837	<2.2 × 10^−16^
HALLMARK_INTERFERON_ALPHA_RESPONSE	97	56	2.0873	<2.2 × 10^−16^
HALLMARK_COMPLEMENT	200	101	2.269	<2.2 × 10^−16^
HALLMARK_IL6_JAK_STAT3_SIGNALING	83	52	2.3259	<2.2 × 10^−16^
HALLMARK_ALLOGRAFT_REJECTION	199	116	2.4589	<2.2 × 10^−16^
HALLMARK_INTERFERON_GAMMA_RESPONSE	200	123	2.608	<2.2 × 10^−16^

## Data Availability

Complete bam, vcf, and maf file datasets are available under controlled access in the public data repository dbGaP Study Accession: phs003404.v1.p1 “Genomic Analysis Of Prostate Tumor Heterogeneity In Metastasis”. Processed read counts from RNAseq and mutational data from WES are available in the Appendix A.

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
