# Peer review of "Integrated Genomic Analysis of Primary Prostate Tumor Foci and Corresponding Lymph Node Metastases Identifies Mutations and Pathways Associated with Metastasis"

_cancers, 2023, doi:10.3390/cancers15235671_

Round 1

Reviewer 1 Report

Comments and Suggestions for Authors

Integrated genomic analysis of primary prostate tumor foci and corresponding lymph node metastases identifies mutations and pathways associated with metastasis

Suggestion: minor revisions

1.      Usually a simple summary of your work is not required before the abstract. However, please ignore this if this was a requirement outlined by the journal/special issue

2.      For figure 3D, it might be useful to label the points using  a single number within the chart and then denoting what each of the number stand for as a key on the side – for clarity

3.      Please add a flowchart in the manuscript to talk through the number of patients at each step (started with 56, 30 of them had metastasis, 7 did not consent to gene analysis etc). This will help in better visualising your availability of samples and your inclusion/exclusion criteria.

4.      For any of the given data sets, did you find any correlation with the age of the 56 patients/165 tissue samples from 43 samples? It would be interesting to see if any of your data did indeed have any correlation the ages of the patients.

5.      Please re-title your conclusions as conclusions and future directions and please add a few more lines towards your future vision of the extensive work accomplished in the manuscript

Comments on the Quality of English Language

Minor editing of English language required

Reviewer 2 Report

Comments and Suggestions for Authors

Reviewer Report on "Integrated genomic analysis of primary prostate tumor foci and corresponding lymph node metastases identifies mutations and pathways associated with metastasis" by Moreno et al.

The authors have made substantial efforts in delineating the genomic landscape of primary prostate tumor and lymph node metastases, yet there are several aspects that require further clarification:

1.     Line 222: The rationale behind selecting the "oxidative phosphorylation" pathway should be elaborated, considering all the other pathways summarized in Table 2.

2.     Line 268: please provide data or figure to support this statement.

3.     Line 275: It's essential to include statistical analysis of OXPHOS scores across all groups to show the significance of differences.

4.     Figure 1 Workflow (Line 233/Figure 1): A workflow or a brief flowchart illustrating how the oxidative phosphorylation pathway was nominated would be highly beneficial.

5.     Statistical Analysis of Fig.1F (Line 281): Similar to the point above, including statistical analysis for Fig.1F is necessary.

6.     Misalignment of Results (Line 243): The results discussed here do not align well with the section title.

7.     Line 341: "most primary tumor foci" please also provide data or figure to support this statement.

8.     Validation of FLT4 and PCDH15 genes (Line 437): It's crucial to explain why FLT4 and PCDH15 were not included for further validation in external datasets despite their novel association with prostate cancer metastases.

9.     WES Analysis Pipeline Details (Line 160): The Methods section on WES Analysis Pipeline needs to be expanded to provide comprehensive information for interested readers.

Round 2

Reviewer 2 Report

Comments and Suggestions for Authors

Thank you for addressing the comments thoroughly.
Based on the revisions and the quality of the research presented, I suggest accepting the manuscript in its latest form.